

# Effects of diet and gizzard muscularity on grit use in domestic chickens

Ryuji Takasaki[1,2] and Yoshitsugu Kobayashi[3]

[1] Department of Natural History Sciences, Hokkaido University, Sapporo, Japan
[2] Faculty of Biosphere-Geosphere Science, Okayama University of Science, Okayama, Japan
[3] Hokkaido University Museum, Hokkaido University, Sapporo, Japan

Corresponding author
Ryuji Takasaki,
rtakasaki@big.ous.ac.jp

## ABSTRACT

The gizzard is the only gastrointestinal organ for mechanical processing in birds. Many birds use grit in the gizzard to enhance mechanical processing efficiency. We conducted an experiment to test the factors that affect chicken grit use in 68 male layer chicks of *Gallus gallus domesticus*, which were divided into two different groups in gizzard muscularity (high and low). Within each muscularity group, two different diets were provided (herbivory and non-herbivory) to test whether diet and gizzard muscularity affect grit characteristics including amount, size, and shape (circularity, roundness, and solidity) at different stages of digestion (ingested grit, grit in gizzard, and excreted grit). All animals ingested more grit than they excreted, possibly because excreted grit was below the detection size limit of 0.5 mm of the present study. The amounts of grit ingested and remained in the gizzard were larger in herbivorous groups, but these groups excreted less grit. Larger, rougher grit was selectively ingested by all chicks, but size preferences were especially pronounced in the herbivorous groups. Grit in the gizzard tended to be larger in herbivorous groups, but the grit in excreta was smaller, whereas the size of excreted grit was larger in groups with less muscular gizzards. Grit in the gizzard was much smoother than the offered and ingested grit, especially in the herbivorous, muscular gizzard groups. Excreted grit in all groups was smoother than the offered grit. These results show that diet affects the characteristics of ingested grit, grit in the gizzard, and excreted grit, whereas gizzard muscularity affects the characteristics of grit in the gizzard and excreted grit. The use of larger sizes and amounts of grit by herbivorous groups may be a response to the needs of digesting hard, coarse materials. The recovered behavioral flexibility of grit use might reflect the omnivorous nature of *Gallus gallus domesticus* and may aid smooth dietary shifts. The results also show that the shape of grit remaining in the gizzard does not reflect the initial shape of ingested grit, in contrast to previously published ideas. Instead, the shape of grit in the gizzard more closely reflects the diet and gizzard muscularity of chicks.

## INTRODUCTION

Digestion, or food processing, is a key phase of animal feeding (*Montuelle & Kane, 2019*). In response to the challenges of digestion, many animal lineages have evolved complex

morphological and physiological adaptations during their evolutionary history. This is particularly true of birds, which have multiple specialized gastrointestinal organs, including a crop for temporal storage of food (*Proctor & Lynch, 1993*), ceca for the conservation of water and/or nitrogen recycling (*DeGolier, Mahoney & Duke, 1999*; *Karasawa, 1989*), and a muscular gizzard for mechanical processing of ingesta (*Moore, 1999*). In the gizzard, strong compression and translational stress mechanically processes ingesta (*Moore, 1998a*). Large food particles are selectively retained in the gizzard until these are ground to a small size (*Hetland, Svihus & Krogdahl, 2003*; *Moore, 1999*). To improve the efficiency of digestion, many birds ingest and retain grit in the gizzard to break down food particles, similar to the use of teeth by non-ruminant mammals (*Fritz et al., 2011*). Some birds even travel long distances to obtain suitable grit in cases where there are insufficient sands or gravels nearby (*McIlhenny, 1932*). Although multiple non-digestive functions such as parasite destruction, relief of hunger, and ballast, have also been proposed for grit use in vertebrates (*Wings, 2007*), grit use is especially common in herbivorous and granivorous birds (*Gionfriddo & Best, 1999*).

The benefits of grit use in domestic chickens have been investigated in poultry science. While grit use is not necessary for survival, previous works generally agree that grit improve digestion efficiency of chickens, especially those fed coarse, less-nutritional food (*Fritz, 1937*; *Hetland, Svihus & Krogdahl, 2003*; *Smith & MacIntyre, 1959*). Additionally, previous works have attempted to identify the best forms and sizes of limestone grit for maximizing egg productivity and quality in layer hens (*Guinotte & Nys, 1991*; *Skřivan et al., 2016*). Multiple studies have also investigated how different grit characteristics (e.g., size, amount, shape) improve nutritional/commercial efficiency in domestic chickens (*Balloun & Phillips, 1956*; *Cooney, 1941*). On the other hand, factors that affect grit use behavior are less understood. This is partly because previous studies are based primarily on grit collected from gizzards (*Best & Gionfriddo, 1991*; *Gionfriddo & Best, 1996*), even though initial grit characteristics can be heavily modified through abrasion in the gizzard (*Buckner, Martin & Peter, 1926*; *Wings & Sander, 2007*).

Here we test whether differences in diet and gizzard muscularity affect the amount, size, and shape of grit at various stages of digestion. This provides insights into how domestic chicks change grit use behavior in response to dietary demands.

## MATERIALS AND METHODS

**Ethics statement:** The experiment in this study was approved by Hokkaido University (Permission number: 16-0023) in Sapporo, Japan, and followed the rules specified in the Hokkaido University manual for implementing animal experimentation.

### Experimental design and managements

A total of 68 one-day-old male layer chicks (*Gallus gallus domesticus*), purchased from a local feed manufacturer, were used in this experiment. This sample size was set based on Hokkaido University regulations, space availability, and several prior experiments conducted on domestic chickens (*Hetland, Svihus & Krogdahl, 2003*; *Van der Meulen,*
*Kwakernaak & Kan, 2008*). Prior to the experiment, the chicks were raised for 3 weeks to produce two groups of differing gizzard muscularity (evaluated as the relative weight of a gizzard to the body mass of the chick). This interval was determined based on the study of *Amerah, Lentle & Ravindran (2007)*, which demonstrated a significant difference in gizzard muscularity at 3 weeks. For 3 weeks, all chicks were fed a commercial starter pellet (Yamaichi Shiryo Co. Ltd., Osaka, Japan). Muscular development of the gizzard (Fig. S1) was enhanced in half of the chicks (34 individuals) by increasing the amount of insoluble fiber in the diet (*Sacranie et al., 2012*). To accomplish this, rice hulls ground to <5 mm (Honda Co. Ltd., Ishikawa, Japan) were mixed (30% by weight) with the starter pellet. The chicks had ad libitum access to feed and water.

The experiment was conducted for 1 week on four groups varying in diet and gizzard muscularity (17 individuals each): a non-herbivorous diet with a muscular gizzard (nH-M), an herbivorous diet with a muscular gizzard (H-M), a non-herbivorous diet with a less-muscular gizzard (nH-lM), and an herbivorous diet with a less-muscular gizzard (H-lM). Herbivorous groups (H-M and H-lM) were fed a mixture of Fabaceae (*Medicago sativa*) and Poaceae (*Phleum pratense*) plants, both purchased from Mapet Corp., Osaka, Japan. Non-herbivorous groups (nH-M and nH-lM) were fed dried fish (*Engraulis japonicas*), purchased from Sakamoto Corp., Nagoya, Japan. The chicks were fed either 100% plant or 100% fish to test the effect of diet on grit use under extreme dietary differences. Plant and dried fish were ground to <5 mm and provided as meal.

During the experiment, all chicks were raised in individual cages with a wire mesh floor. Room temperature was maintained between 28 °C and 30 °C. Lighting was controlled to provide a 12-h light/dark cycle. All of the chicks were given ad libitum access to a total of 24 g of grit per chick, which was provided separately from the feed. Feces were collected on the last day of the experiment to evaluate the characteristics of excreted grit. All chicks were weighed and then euthanized by cervical dislocation at the end of the experiment, following Hokkaido University regulations. The gizzard was removed from each of the carcasses and weighed after removing stomach contents.

## Terminology

We use the term offered grit to refer to all of the stones which were given with ad libitum access to the chicks (Fig. 1A). Uningested grit refers to the grains that were not ingested by the chicks from the offered grit by the end of the experiment (i.e., the remainder of the offered grit). Ingested grit refers to the grains that were selected and swallowed by the chicks from the offered grit during the experiment; this was measured as the difference between the offered and the uningested grit fractions. Grit in the gizzard refers to the particles remaining in the gizzards of the chicks after euthanasia. Excreted grit refers to the particles excreted with the feces on the last day of the experiment. Initial gizzard muscularity refers to the gizzard muscularity of chicks at the start of the experiment. "Rough" is used to describe grit with relatively low circularity, roundness, and/or solidity, and "smooth" is used to describe grit with relatively high shape index.
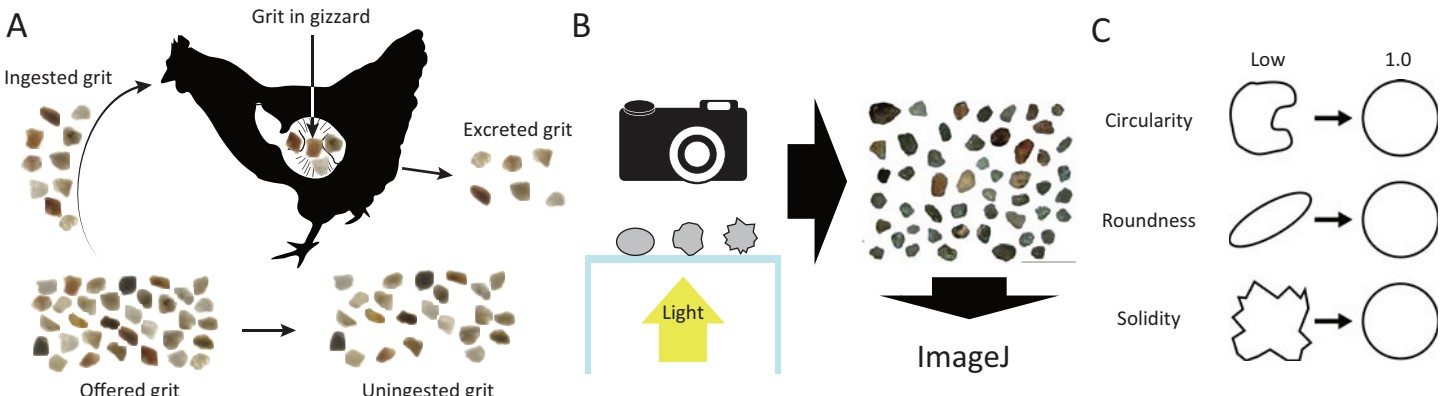

**Figure 1** **Visualized experimental design.** (A) Terminology regarding grit treated in this study. (B) Grit shape evaluation procedure. (C) Schematic explanation of grit shape indexes.

## Grit characteristics

All of the grit used in the experiment were commercial quartzite stones. The amount, size, and shape of grit at different stages of digestion (i.e., offered grit, uningested grit, grit in the gizzard, and excreted grit) were evaluated. The amount of grit was weighed in grams. Size and shape were evaluated quantitatively in ImageJ using the menu command Analyze > Analyze Particles (*Schneider, Rasband & Eliceiri, 2012*). To obtain the images for these analyses, grains were manually separated and backlit to obtain clear outlines (Figs. 1B and 2). All images were taken manually. The minor axis of a particle was used to measure grit size (in millimeters), and the minimum size threshold was 0.5 mm (i.e., grit particles <0.5 mm were not analyzed). Circularity, roundness, and solidity were employed as grit shape indices (*Schneider, Rasband & Eliceiri, 2012*). Circularity was calculated as four times the product of π and area, divided by the square of the perimeter. Roundness was taken from the inverse of the aspect ratio. Solidity was calculated as the area of a grain divided by the area of the convex hull (Fig. 1C). The quantitative grit shape evaluation methods used here are not direct equivalents of the qualitative grit shape evaluation proposed by *Best & Gionfriddo (1991)*. For example, the quantitative methods are less powerful in detecting corner sharpness, whereas the qualitative methods do not take aspect ratio into account. Despite the differences, both methods adequately evaluate grit shape and this study adopts the quantitative method for scientific objectivity.

### *Offered grit*

The amount, size, and shape of ingested grit were inferred by comparing the characteristics of offered and uningested grit. These measurements were performed on a group-by-group basis. To test grit size preference of the chicks, the size distribution of offered grit was controlled in advance. Grit was classified into six different size classes by dry sieving using mesh sieves (Sanpo Corp.; 0.5–1.0 mm, 1.0–1.4 mm, 1.4–1.7 mm, 1.7–2.0 mm, 2.0–2.8 mm, 2.8–3.35 mm). Four grams of grit from each size class were supplied in the mixture. Before the experiment, minor axis, circularity, roundness, and solidity of 500 randomly chosen grit from each grit size class were evaluated using ImageJ (Fig. S2). After the experiment, uningested grit was collected and sieved again into the six size
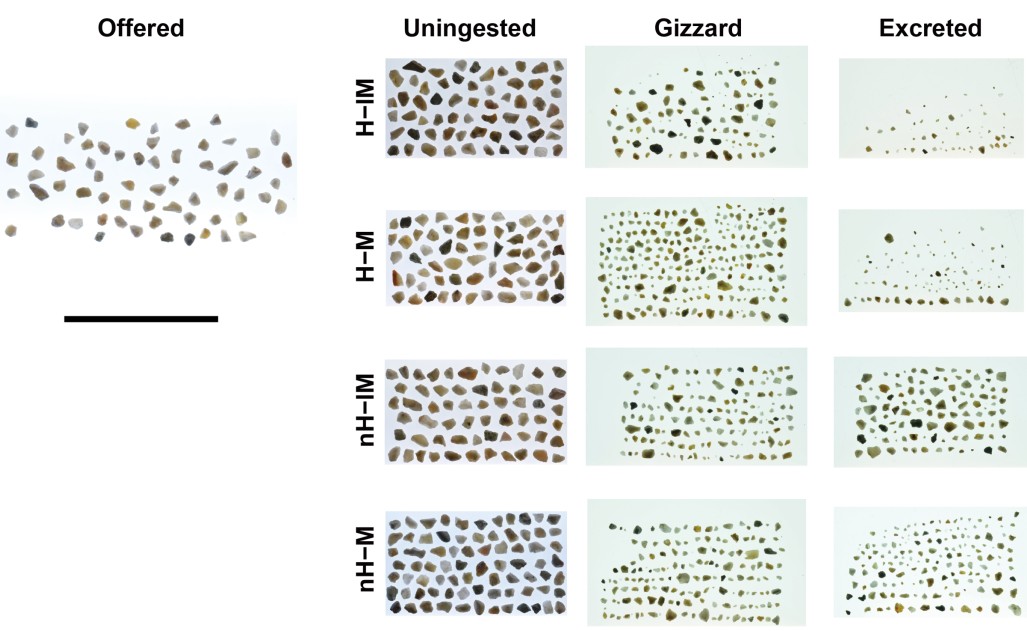

**Figure 2 Representatives offered grit, uningested grit, grit in the gizzard, and excreted grit.** Note that the grit shown here are only few representatives from hundreds of grit therefore they may not reflect true size/shape distribution. Abbreviations: H-lM, herbivorous diet with a less-muscular gizzard; H-M, herbivorous diet with a muscular gizzard; nH-lM, non-herbivorous diet with a less-muscular gizzard; nH-M, non-herbivorous diet with a muscular gizzard. Scale = 50 mm.

classes. Five-hundred uningested grit were randomly sampled from each size classes and their size and shape indices were evaluated (Fig. S2). Because the weight of the 500 randomly-selected grit of each size category was not determined, the total number of grit per size class could not be reconstructed, and therefore the average particle size of offered and uningested grit is unavailable. To test size preferences, the amount of ingested grit in each size class was measured as the difference between the weights of the uningested grit and the weights of the offered grit, by size class (4 g each). Average shape indices of the uningested grit were compared to those of the offered grit to test if there was any shape preference in the ingested grit. The amounts, sizes, and shapes of the uningested grit were then compared among different dietary and gizzard muscularity groups.

### Grit in the gizzard

Grit in the gizzard was separated from other stomach contents using a flotation method (decantation) modified from *Itani (2015)*. Stomach contents were soaked in water in a beaker overnight. Gizzard digesta was stirred and then low-density floating food particles were gently discarded. This procedure was repeated until only grit remained in the beaker. Grit smaller than 0.5 mm were collected if possible, but were removed from the analyses. The total amount of grit was weighed, and then the size and shape of all grit particles in the gizzard were analyzed using ImageJ (Fig. S2). The amounts, sizes, and shapes of the gizzard grit were compared among different dietary and gizzard muscularity groups.

### Excreted grit

Excreted grit was collected and evaluated on a group basis. Grit particles were separated from fecal particles using the same decantation method as for separating gizzard grit. Grit smaller than 0.5 mm were collected if possible, but were removed from the analyses. The total amount of grit was weighed and the sizes and shapes of all excreted grit were analyzed using ImageJ (Fig. S2). The amounts, sizes, and shapes of the excreted grit were evaluated and compared with those of the gizzard grit. Grit characteristics were also compared among different diet and gizzard muscularity groups. Because excreted grit was only collected on the last day of the experiment, and because it was collected per group instead of per individual, the amount and size of excreted grit per individual is unavailable in the present study.

## Statistical analyses

All statistical analyses were conducted using R software package (R Core Team, 2019). Because some of the datasets did not have a normal distribution, non-parametric analyses were conducted throughout. Ordinal logistic regressions were conducted as a non-parametric equivalent of two-way ANOVA to test the effects of gizzard muscularity, diet, and their interactions for body mass, gizzard mass, and grit features (size, amount, and shape), using the R package MASS (Venables & Ripley, 2013). The Steel-Dwass method was used for post-hoc tests. Correlations between shape indices of gizzard grit and gizzard muscularity at the end of the experiment were tested using Spearman's rank correlation. Grit amount was corrected using chick body mass and grit size was corrected using the cubic root of chick body mass in the analyses. Average chick body mass and grit amount were compared on an individual basis (the average of individuals per group), whereas average grit size, circularity, roundness, and solidity were compared on grit basis (the average of the grits in each group).

Chicks that were euthanized before the end of the experiment following Hokkaido University regulations were excluded from the analyses. The data analyzed are provided as Supplemental Information (Data S1–S6).

## RESULTS

During the experiment, two chicks from the non-herbivorous, muscular gizzard group (nH-M), one chick from the non-herbivorous, less-muscular gizzard group (nH-lM), five chicks from the herbivorous, muscular gizzard group (H-M), and two chicks from the herbivorous, less-muscular gizzard group (H-lM) were euthanized before the end of the experiment due to a sudden drop in body mass, following Hokkaido University regulations. Therefore, the analyses were performed on a total of 58 chicks.

The average body mass of the chicks at the end of the experiment was affected by both diet and initial gizzard muscularity, as well as the interaction of these factors (Table S1). Body masses were higher in the non-herbivorous groups than in the herbivorous groups (nH-M > H-M, nH-lM > H-lM; Table 1; Table S2). This difference in body mass was significant only between the groups with high initial gizzard muscularity. Average gizzard muscularity at the end of the experiment was affected by diet and initial

**Table 1 Average values of chick status, grit amount, and grit size.** Note that average sizes of offered and remained grit cannot be calculated, but assumed from the amount of stones ingested per size class. See Fig. 3 and Table S3 for detail. Abbreviations: H-lM, herbivorous diet with a less-muscular gizzard; H-M, herbivorous diet with a muscular gizzard; nH-lM, non-herbivorous diet with a less-muscular gizzard; nH-M, non-herbivorous diet with a muscular gizzard.

| | Chick (g) | | | | Grit amount (g) | | | | | | Grit size (mm)[**] | | | | |
|---|---|---|---|---|---|---|---|---|---|---|---|---|---|---|---|
| | Body mass | SD | Gizzard mass | SD | Ingested | SD | Gizzard | SD | Feces | SD | Uningested | Gizzard | SD | Feces | SD |
| **Raw values** | | | | | | | | | | | | | | | |
| H-lM | 182.736[a] | 33.769 | 7.531 | 1.484 | 4.852[ab] | 3.484 | 1.305 | 0.525 | 0.09[*] | NA | NA | 1.884[ab] | 0.340 | 0.960[a] | 0.377 |
| H-M | 153.403[bc] | 26.688 | 7.863 | 1.483 | 3.657 | 2.572 | 1.118 | 0.498 | 0.26[*] | NA | NA | 1.743[acd] | 0.355 | 0.975[bc] | 0.505 |
| nH-lM | 225.163[b] | 56.007 | 6.636[a] | 1.501 | 1.879[a] | 1.252 | 0.781 | 0.634 | 1.34[*] | NA | NA | 2.003[ce] | 0.413 | 1.273[abd] | 0.556 |
| nH-M | 250.977[ac] | 44.363 | 8.205[a] | 1.227 | 2.318[b] | 1.810 | 0.919 | 0.784 | 0.83[*] | NA | NA | 1.883[bde] | 0.512 | 1.138[cd] | 0.500 |
| **Relative to body mass** | | | | | | | | | | | | | | | |
| H-lM | NA | NA | 0.041[abc] | 0.041 | 0.026[ab] | 0.004 | 0.007[ab] | 0.003 | NA | NA | NA | 0.332[ab] | 0.055 | 0.169[a] | 0.066 |
| H-M | NA | NA | 0.051[ade] | 0.051 | 0.022[cd] | 0.006 | 0.007[cd] | 0.003 | NA | NA | NA | 0.326[c] | 0.065 | 0.182[b] | 0.094 |
| nH-lM | NA | NA | 0.030[bd] | 0.03 | 0.009[ac] | 0.005 | 0.003[ac] | 0.003 | NA | NA | NA | 0.329[ad] | 0.067 | 0.209[abc] | 0.091 |
| nH-M | NA | NA | 0.033[ce] | 0.033 | 0.009[bd] | 0.004 | 0.003[bd] | 0.003 | NA | NA | NA | 0.298[bcd] | 0.087 | 0.180[c] | 0.079 |

Notes:
[*] Total amount excreted on the last day of the experiment, evaluated on group basis.
[**] Grit size are averaged and compared by grit basis instead of individual basis, unlike chick status and grit amount.
[a–e] Means within a column sharing a common superscript differ significantly at $p < 0.05$.

gizzard muscularity (Table S1). The average gizzard muscularity was significantly higher in herbivorous groups than in non-herbivorous groups (H-M > nH-M, H-lM > nH-lM; Table S2). While the differences in initial gizzard muscularity remained significant between the herbivorous groups at the end of the experiment (H-M > H-lM), the differences were insignificant between the non-herbivorous groups. This result is likely to reflect a rapid change in gizzard muscularity associated with diet change.

## Grit amount

The experiment demonstrated that amount of ingested grit per chick throughout the experiment was approximately 3 g in average (1.7% of body mass), whereas the amount of grit in the gizzard was approximately 1 g in average (0.5% of body mass), suggesting that about two-thirds of the ingested stones were excreted during the experiment (Table 1). The amount of excreted grit on the last day of the experiment was 0.045 g per chick, on average. Diet affected the average amount of ingested grit and grit in the gizzard, both in total and relative to body mass (Table S1). The post-hoc tests showed that herbivorous groups ingested significantly more grit relative to their body mass (H-M > nH-M, H-lM > nH-lM; Table S2). The amount of grit in the gizzard relative to body mass was also greater in herbivorous groups (H-M > nH-M, H-lM > nH-lM; Table S2). No significant difference in the amount of ingested grit and grit in the gizzard was detected between groups with differing initial gizzard muscularity. The amount of excreted grit on the last day of the experiment was larger in non-herbivorous groups than in herbivorous groups in total
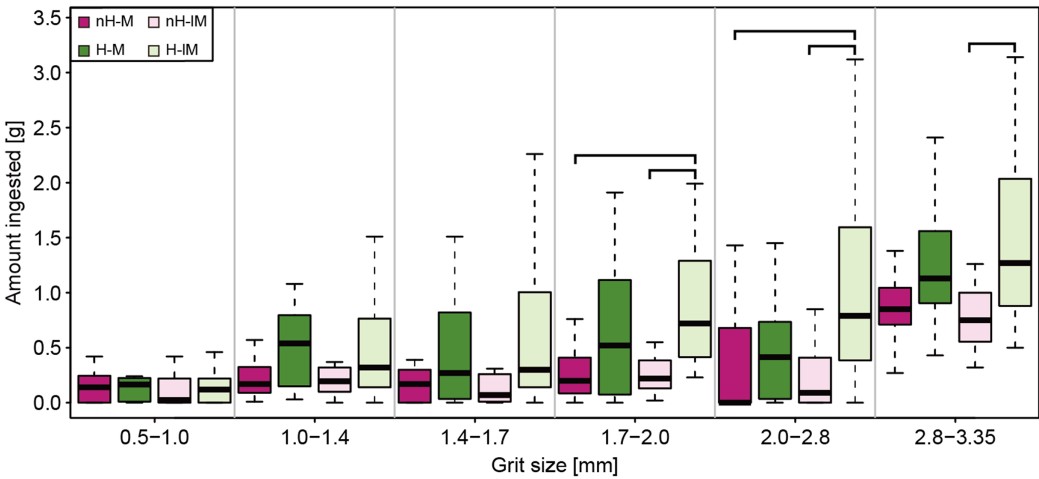

**Figure 3 Boxplots showing the amount of ingested grit by experimental groups, shown per grit size categories.** The brackets represent significant differences at $p < 0.05$. Abbreviations: H-lM, herbivorous diet with a less-muscular gizzard; H-M, herbivorous diet with a muscular gizzard; nH-lM, non-herbivorous diet with a less-muscular gizzard; nH-M, non-herbivorous diet with a muscular gizzard.

weights (nH-M: 0.83 g, H-M: 0.26 g, nH-lM: 1.34 g, H-lM: 0.09 g), although no statistical test is available to test its significance ($n = 1$ per group).

## Grit size

Large grit (>2.8 mm) were generally ingested more than smaller grit (Fig. 3; Table S3). The average size of grit in the gizzard was about 1.84 mm and that of the excreted grit was 1.09 mm (Table 1). Diet affected the ingestion of grit larger than 1.4 mm (Table S4). Post-hoc tests show that this difference was significant between herbivorous and non-herbivorous groups of the less-muscular gizzard group (H-lM > nH-lM; Table S5). Diet and the interaction of gizzard muscularity with diet affected the average size of grit in gizzard relative to body mass (Table S1). While the average absolute sizes of the grit in the gizzard were larger in non-herbivorous groups, the average sizes of grit in the gizzard relative to body mass were significantly larger in herbivorous groups (Table 1; Table S2). Within non-herbivorous groups, the less-muscular gizzard group contained significantly larger grit in the gizzard relative to their body mass than the muscular gizzard group (nH-lM > nH-M; Table 1; Table S2). The average size of excreted grit was affected by diet and the interaction of diet and gizzard muscularity (Table S1). Within the less-muscular gizzard group, excreted grit was larger in the non-herbivorous chicks than in the herbivorous chicks (nH-lM > H-lM; Table S2). Furthermore, excreted grit was larger in the non-herbivorous, less-muscular group compared to the non-herbivorous, muscular group (nH-lM > nH-M; Table S2), both in absolute and corrected values.

## Grit shape

Circularity, roundness, and solidity of the uningested grit was higher than that of the offered grit (Figs. 4A–4C; Table 2), suggesting that ingested grit had low circularity, roundness, and solidity. This trend in circularity and solidity is generally significant for grit

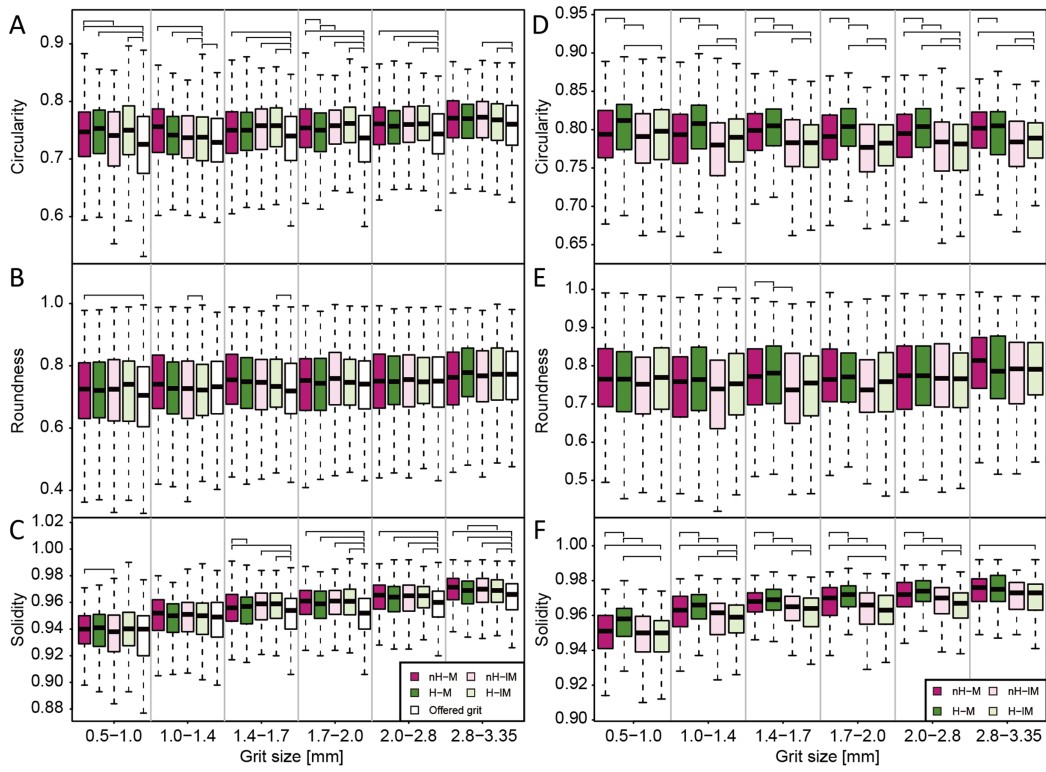

**Figure 4  Boxplots comparing grit shapes.** (A–C) Boxplots comparing shapes of the initial grits and the remained grits by each experimental group, shown per grit size categories. (D–F) Boxplots comparing shapes of the grit in the gizzard by the experimental groups, shown per grit size categories. The brackets represent significant differences at $p < 0.05$. Abbreviations: H-lM, herbivorous diet with a less-muscular gizzard; H-M, herbivorous diet with a muscular gizzard; nH-lM, non-herbivorous diet with a less-muscular gizzard; nH-M, non-herbivorous diet with a muscular gizzard.

larger than 1.4 mm (Table S6). Grit in the gizzard had higher circularity, roundness, and solidity than both the offered and uningested grit (Table 2). This trend in circularity and solidity was significant for all size classes and the trend in roundness was generally significant for herbivorous groups (Table S7). The circularity and solidity of excreted grit were higher than those of grit in the gizzard (Table 2), although this trend is only significant for solidity (Table S8). The shape indices of excreted grit were higher than those of the offered grit (generally significant for circularity and roundness; Table S8).

Neither diet nor gizzard muscularity strongly affected shape indices of the uningested grit (Tables S9 and S10). Diet and gizzard muscularity did affect the circularity of the grit in the gizzard of nearly all grit size classes, while diet, gizzard muscularity, and their interaction affected the solidity of grit in gizzard (Figs. 4D–4F; Table S11). On the other hand, the roundness of grit in the gizzard was affected only by diet in size classes 1.0–2.0 mm. Post-hoc tests show that grit in the gizzard of herbivorous groups was significantly higher in circularity and solidity for most size classes (H-M > nH-M, H-lM > nH-lM; Figs. 4D–4F; Table S12). Circularity and solidity of the grit in the gizzard were also higher in the herbivorous, muscular gizzard group than in the herbivorous, less-muscular

| | Offered | SD | Uningested | SD | Gizzard | SD | Feces | SD |
|---|---|---|---|---|---|---|---|---|
| **Table 2** Average values of shape indexes. | | | | | | | | |
| | Circularity | | | | | | | |
| H-lM | 0.734 | 0.060 | 0.749 | 0.057 | 0.789 | 0.045 | 0.773 | 0.045 |
| H-M | 0.734 | 0.060 | 0.746 | 0.055 | 0.798 | 0.045 | 0.783 | 0.045 |
| nH-lM | 0.734 | 0.060 | 0.747 | 0.058 | 0.776 | 0.049 | 0.774 | 0.049 |
| nH-M | 0.734 | 0.060 | 0.752 | 0.054 | 0.780 | 0.051 | 0.777 | 0.051 |
| | Roundness | | | | | | | |
| H-lM | 0.732 | 0.121 | 0.742 | 0.121 | 0.762 | 0.108 | 0.760 | 0.119 |
| H-M | 0.732 | 0.121 | 0.740 | 0.119 | 0.764 | 0.107 | 0.763 | 0.099 |
| nH-lM | 0.732 | 0.121 | 0.738 | 0.122 | 0.745 | 0.113 | 0.762 | 0.116 |
| nH-M | 0.732 | 0.121 | 0.741 | 0.116 | 0.752 | 0.110 | 0.753 | 0.113 |
| | Solidity | | | | | | | |
| H-lM | 0.950 | 0.021 | 0.954 | 0.021 | 0.961 | 0.016 | 0.947 | 0.020 |
| H-M | 0.950 | 0.021 | 0.953 | 0.020 | 0.964 | 0.015 | 0.953 | 0.016 |
| nH-lM | 0.950 | 0.021 | 0.954 | 0.021 | 0.959 | 0.016 | 0.953 | 0.016 |
| nH-M | 0.950 | 0.021 | 0.955 | 0.019 | 0.954 | 0.017 | 0.951 | 0.017 |

Note:
H-lM, herbivorous diet with a less-muscular gizzard; H-M, herbivorous diet with a muscular gizzard; nH-lM, non-herbivorous diet with a less-muscular gizzard; nH-M, non-herbivorous diet with a muscular gizzard.

gizzard group (H-M > H-lM; Table S12). Circularity, roundness, and solidity were correlated with gizzard muscularity ($p < 0.05$). The solidity of excreted grit is inferred to be affected by diet and the interaction of gizzard muscularity and diet (Table S13), although the difference was undetected in post-hoc tests (Table S14).

## DISCUSSION

### Grit amount

The larger amounts of ingested grit and grit in the gizzard in herbivorous groups (H-lM > nH-lM; Tables S2 and S3) are concordant with previous studies (see a comprehensive review by *Gionfriddo & Best (1999)*). The larger amount of excreted grit in the non-herbivorous groups (nH-M > H-M, nH-lM > H-lM) might suggest that the large amount of grit in the gizzard in herbivorous groups was driven by higher ingestion rates of grit in combination with limited grit excretions. Retaining a larger amount of grit in the gizzard is likely to benefit herbivorous groups by helping to break down tough plant fibers, because a larger amount of grit in the gizzard improves digestive performance in domestic chickens (*Bale-Therik, Sabuna & Jusoff, 2012*), as long as the amount is not excessive (*Moore, 1998b*).

Assuming equivalent intake and excretion of grit, the total weight of grit theoretically excreted throughout the experiment would be 20.99, 30.46, 17.58, and 53.22 g in groups nH-M, H-M, nH-lM, and H-lM, respectively. However, these values are higher than the measured amounts of grit excreted. Reasonable explanations for this are that the chicks excreted more grit on days prior to fecal collection, or that a high proportion of the excreted grains were less than 0.5 mm in size and were therefore undetected. While both

are likely, the latter explanation suggests that up to 20.36, 28.64, 8.20, and 47.42 g of grit in groups nH-M, H-M, nH-lM, and H-lM, respectively, were abraded to less than 0.5 mm. The larger amount of lost grit in herbivorous groups (H-M > nH-M, H-lM > nH-lM) might indicate a more thorough particle size reduction of grit due to more extensive use in the herbivorous groups. This would result in a larger fraction of the grit being excreted at sizes below the detection limit of the present study. The fact that, on average, larger grains were excreted by the non-herbivorous group with less muscular gizzards supports this latter interpretation. However, a more specific experiment with specific daily records on the amount of excreted grit would be required to make any further conclusions.

## Grit size

Because the size of grit in the gizzard is unlikely to affect digestion efficiency in domestic chickens (Smith, 1960), and larger grit may even be associated with lower digestion efficiency (Moore, 1998c), selective ingestion of larger grit in all groups (Table S3) may simply reflect the ease of picking larger grains. The smaller size of the excreted grit than the grit in the gizzard (Table S8) in all groups suggests that size is one of the primary factors that determines which grains are to be excreted in domestic chickens. While the excretion of small grit is concordant with a trend in domestic chickens (Smith, 1960), it contrasts with a trend reported in House Sparrows Passer domesticus (Gionfriddo & Best, 1995). Therefore, the responses in the size of excreted grit may vary taxonomically. The larger sizes of ingested grit and grit in the gizzard (relative to body mass) in herbivorous groups (Fig. 3; Table 1; Tables S2 and S5) agree with previous works (Gionfriddo & Best, 1999; Hoskin, Guthrie & Hoffman, 1970; May & Braun, 1973; Norris, Norris & Steen, 1975; Soler, Soler & Martinez, 1993; Thomas, Owen & Richards, 1977). The large size of excreted grit in the non-herbivorous, less-muscular gizzard group (Table 1) suggests that this group either could not retain grit in the gizzard or did not use the grit as extensively, thus failing to reduce its size. Both possibilities are consistent with the large amount of excreted grit measuring greater than 0.5 mm in this group.

## Grit shape and abrasions

The higher average roughness of the ingested grit (having higher shape indexes) than the offered grit (Figs. 4A–4C; Table S6) is consistent with previous knowledge in domestic chickens (Smith, 1960) as well as in House Sparrows Passer domesticus and the Northern Bobwhite Colinus virginianus (Best & Gionfriddo, 1994). Moore (1998c) showed that rough grit may increase comminution efficiency. Therefore, in each group, actively ingesting rough grit may reflect a congenital behavior for better digestion efficiency, although further investigation into the impacts of grit shape on comminution efficiency is necessary. That the grit in the gizzard is smoother on average than the offered grit (Table S7) contrasts the selective ingestion of rougher grit. Because the excreted grit was also smoother than the offered grit (Table S8), it is most likely that the grit in the gizzard was severely abraded inside the gizzard (Wings & Sander, 2007). Severe grit abrasion and the

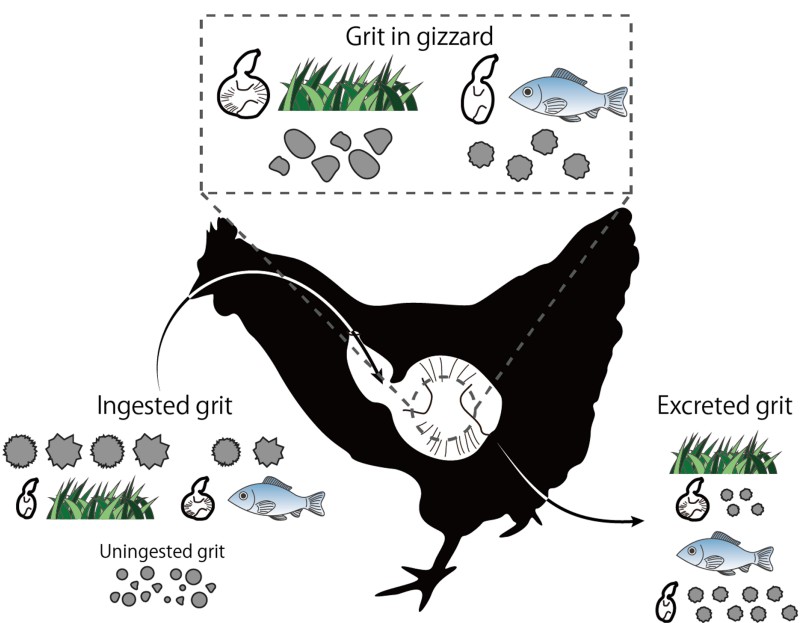

**Figure 5 Schematic summary of the results of this experiment.**

associated grain size reduction are both concordant with a large amount of lost grit (see above).

The dominance of smooth grit in the gizzard of herbivorous groups, as well as in the more muscular gizzard groups (H-M > nH-M, H-lM > nH-lM, H-M > H-lM; Figs. 4D–4F; Table S12), strongly suggests that diet and gizzard muscularity affect the degree of abrasion of grit in the gizzard. Because dietary structures significantly affect gizzard muscularity in birds, including domestic chickens (*Dekinga et al., 2001*; *Hetland, Svihus & Krogdahl, 2003*; *Sacranie et al., 2012*), gizzard muscularity may be a primary factor in determining the degree of abrasion of grit in the gizzard. Correlations between gizzard muscularity and shape indices of grit in the gizzard are also consistent with this interpretation. Therefore, the shapes of particles in the gizzard are unlikely to fully reflect grit selection patterns in domestic chickens, in contrast to previously published concepts (*Best & Gionfriddo, 1991*; *Gionfriddo & Best, 1996*). Instead, our experiment suggests that the differences in the shapes of grains in the gizzard more strongly reflect differences in diets and gizzard muscularity. At the same time, however, it should be noted that in cases where birds can only access smoother stones for use as grit in natural conditions, grit might be expected to be less abraded in the gizzard and may thus more strongly reflect the original shape. Investigations using broader taxonomic datasets based on wild birds would be expected to provide further insights.

## Chick grit use behaviors

This study is the first attempt to examine whether diet and gizzard muscularity affect chicken grit use behaviors throughout ingestion, retention, and excretion. This experiment strongly suggests that, under the experimental conditions used here, grit characteristics

were primarily affected by diet and secondarily by the muscularity of the gizzard (Fig. 5; Tables 1 and 2). The flexibility of grit use in response to the needs of digesting tough, coarse ingesta may reflect the omnivorous nature of *Gallus gallus domesticus* and might facilitate easy shifts between herbivorous and carnivorous diets. Because numerous other bird species are known to be omnivorous and experience seasonal diet shifts (*Del Hoyo, Elliott & Christie, 2005*), flexibility in the use of grit in the gizzard may not be limited to domestic chickens. Rather, it might be common, and may support the wide dietary range of omnivorous birds, together with phenotypic flexibility of the gizzard (*Dekinga et al., 2001*; *Starck, 1999*; *Van Gils et al., 2005*). Further studies on other birds are required to test this hypothesis.

## CONCLUSION

This experiment on chick grit use behaviors demonstrated that diet and gizzard muscularity affect the size, amount, and the shape of ingested and excreted grit. It also revealed that grit in the gizzard was greatly modified through abrasion; therefore, grit did not retain its original sizes nor shapes upon ingestion. Instead, gizzard grit shapes reflected gizzard activity, as determined by chick diet and gizzard muscularity: roughest in the chicks with less-muscular gizzards on a non-herbivorous diet and smoothest in the chicks with muscular gizzards on a herbivorous diet. Selective ingestion of rough grit regardless of diet and gizzard muscularity is likely an innate selective behavior. On the other hand, the ingestion of a larger amount of grit by the herbivorous groups may be a behavioral adaptation to ensure an adequate supply of grit as it is abraded during the course of the grinding action of coarse ingesta. The flexibility of grit use by individuals of a different diet, which is expected to reflect gizzard activity, may reflect the omnivorous nature of chickens, and possibly facilitate their seasonal diet shifts in nature.

## ACKNOWLEDGEMENTS

We thank Y. Deguchi and F. Kobari for allowing access to the experimental rooms and loan of experimental equipment, and T. Matsushima to detailed advice on chick management and experimental designs. The first author greatly appreciates T. Tanaka for helping out conducting the experiment. We also thank K. Inada, M. Iijima, Junki Yoshida, Chinzorig Tsogtbaatar, and A.R. Fiorillo for their helpful discussions and valuable insights. We appleciate G. Funston for reviewing and editing the language, clarity, and style of the manuscript.

### Funding

This work was supported by Grant-in-Aid for JSPS Research Fellow Grant Number 17J06410. The funders had no role in study design, data collection and analysis, decision to publish, or preparation of the manuscript.

## Grant Disclosures

The following grant information was disclosed by the authors:
Grant-in-Aid for JSPS Research Fellow: 17J06410.

## Competing Interests

The authors declare that they have no competing interests.

## Author Contributions

- Ryuji Takasaki conceived and designed the experiments, performed the experiments, analyzed the data, prepared figures and/or tables, authored or reviewed drafts of the paper, and approved the final draft.
- Yoshitsugu Kobayashi conceived and designed the experiments, authored or reviewed drafts of the paper, and approved the final draft.

## Animal Ethics

The following information was supplied relating to ethical approvals (i.e., approving body and any reference numbers):

Institutional Animal Care and Use Committee of National University Corporation Hokkaido University provided full approval for this research (16-0023).

## Data Availability

The raw data are available in the Supplemental Files.

## Supplemental Information

Supplemental information for this article can be found online at http://dx.doi.org/10.7717/peerj.10277#supplemental-information.

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
