# Peer review of "Effects of diet and gizzard muscularity on grit use in domestic chickens"

_PeerJ, doi:10.7717/peerj.10277_

## Round 0.1 · original submission · Major Revisions

Dear Authors,

Please read the comments of the reviewers carefully and try to use their recommendations in your revision. The reviewers will check your revision.

Kind regards

Michael Wink
AE

·

Basic reporting

New literature on gastroliths and grit use in birds is rare and every new paper is a very welcomed addition to our yet limited understanding of this matter.
The manuscript by Takasaki & Kobayashi reports a very interesting but relatively short-termed experiment with grit use in chicken, focusing on gastroliths development depended on diet and muscularity of the gizzard.
I found the language of the manuscript sometimes cumbersome. For example, the presentation of results in the abstract could be shorter and written more clearly, with a focus on just the significant results.
There are three major problems I have with the manuscript as it stands:
1) there is virtually no detailed information about the grit used in this study (except its shape)
2) these is not enough relevant literature regarding grit use in domestic chicken cited and discussed
3) the visual presentation of data is poor, it could be very much improved with more easily accessible figures and diagrams
If these issues are remedied, the paper would be a valuable addition to research on bird digestion, gizzards, and gastroliths.

Experimental design

My main issue is with the short duration of the experiment and the validity of its results. See detailed comments below.

As for the animal usage: I consider the chicken experiments necessary and ethical, the necessary approval statements have been provided.

Validity of the findings

See detailed comments below.

Additional comments

Here are some specific comments referring to the line of the manuscript:

14 – To my knowledge there is no plural for grit (the term always incorporates several small stones). “Grits” is a term rather used for groats.
20 – there are well-established and well-defined terms for clast shapes in the geological literature. what do you mean by solidity? hardness?
22 – larger than what? are these changes significant?
39/40 – after reading the complete manuscript, I am not sure you can make such a statement because your experiment with grit only lasted a week!
51 – perhaps mention the other relevant functions of grit (reviewed in: Wings O. A review of gastrolith function with implications for fossil vertebrates and a revised classification. Acta Palaeontologica Polonica. 2007;52(1):1-16.)
83 – I am not sure that three weeks will make such a big difference in the development of (or lack of) gizzard muscles. Please explain why you did not choose a longer period for this experiment? Anyhow, there often is a high plasticity of gizzard muscles in birds.
106 – What was the lithological composition of grit used? This is perhaps the most important factor for development of grit shape changes! Was it a mixture of rock types with different hardnesses? Or was it just quartz or a similarily hard rock type? With hard lithologies, the changes on the grit would only have been minimal after one week (see Wings & Sander 2007). Another example: if you had some limestone pieces in your grit (often incorporated in the commercially used grit for poultry), these clasts would simply dissolve in the stomach juices…
what was the shape variety in initial grit?
112 – Ok, here you define the term solidity. Please provide references for this use in the literature.

170 – What was the reason that chicks had to be euthanized prematurely?
175 – This makes me wonder if the chicks would have chosen even larger grit particles if you would have supplied them. On what basis did you choose the upper grit size limit?

General comments for the results: it would be good if you could illustrate the original shape variety of your initial grit in a diagram. you could also illustrate all other grit samples (ingested, excreted, remaining) in the same way. perhaps with area diagramms? or scatter plots? this would make your main results far easier to compare and comprehend.
Also, it would be helpful to have photographs of the different grit samples. at least in the supplements. also, I would be interested to see photographs of a typical gizzard from each group.

259 – what do you mean with “dull? less polished? or less sharp? be specific
261 – of course gastroliths are heavily abraded in the gizzard. this has been documented by several studies see Wings & Sander 2007
269 – grit abrasion is also strongly dependent on the lithology
273 – this is very strongly dependent on the duration gastroliths remain in the gizzard
278 – I am not sure this really is the first attempt. There is a vast number of papers about grit use in commercial chicken out there. Here are just some from my database which could be incorporated in your discussion:
Balloun, S. L., and Phillips, R. E.: Grit feeding affects growth and feed utilization of chicks and egg production of laying hens, Poult. Sci., 35, 566-569, 1956.
Bgatov, V. I., Motovilov, K. Y., and Speshilova, M. A.: Фyнкции пpиpoдныx минepaлoв в oбмeнныx пpoцeccax ceльcкoxoзяйcтвeннoй птицы [Functions of natural minerals in the metabolic processes of poultry], C.-x. биoлoгия [Agricultural Biology], 7, 98-102, 1987.
Buckner, G. D., and Martin, J. H.: The function of grit in the gizzard of the chicken, Poult. Sci., 1, 108-113, 1922.
Combs, G. F., G.L., R., and Nicholson, J. L.: Studies on the evaluation of insoluble grit for broilers, Maryland Agr. Exp. Sta. Misc.Publ., 210, 1954.
Cooney, W. T.: Influence of various grits on battery-raised broilers, Station circular / Agricultural Experiment Station / Oregon State Agricultural College, 139, 1941.
Curtis, M. R.: On the ability of chickens to digest small pieces of aluminum, Ann. Rept. Maine Agric. Exper. Sta. Bull, 221, 314-318, 1913.
Ferber, K. E., and Brüggemann, H.: Die Zugabe von Kalksteingrit und Flintgrit zum Futter bei der Jungmast von Hähnchen, Archiv für Geflügelkunde, 7, 363-368, 1933.
Heuser, G. F., and Noms, L. C.: Calcite grit and granite grit as supplements to a chick starting ration, Poult. Sci., 25, 195-198, 1946.
Kaupp, B. F., and Ivey, J. E.: Digestive coefficients of poultry feeds and the rapidity of digestion and fate of grit in the fowl, North Carolina Agr. Exp. Sta. Tech. Bul., 22, 1923.
Platt, C. S., and Stephenson, A. B.: The influence of commercial limestone and mica grits upon growth, feed utilization, and gizzard measurements of chicks, New Jersey Agr. Exp. Sta. Bul., 587, 1935.
Rau, G. J., and Platt, C. S.: The effect of size of limestone grit particles in poultry rations, Poult. Sci., 28, 232-235, 1949.
Scott, M. L., and Heuser, G. F.: The value of grit for chickens and turkeys, Journal of Poultry Science, 36, 276-283, 1957.
Tepper, A. E. R., Durgin, R. C., and Botorff, C. A.: Fine versus coarse grit as a feed ingredient for poultry, New Hampshire Agr. Exp. Sta. Circ., 56, 1939.
Titus, H. W., and Fritz, J. C.: The Scientific Feeding of Chickens, 5th ed., Interstate, Danville, Ill.,, 336 pp., 1971.

·

Basic reporting

The manuscript describes a very interesting experiment that is of high biological interest in my view.
The language is not acceptable, and requires extensive revision, including the involvement of a professional in avian biology/nutrition in my view. I made extremely detailed language corrections in the abstract in the attached pdf, and a lot of the detailed criticism can be transferred to the whole manuscript from there. Examples are the use of plural for grit ("grits"), the use of the word "regulation" when a word like "response" or "effect" would be adequate, or "remained grit" for leftovers. Correcting the language of the whole article is beyond the scope of work a reviewer can do.
The literature references are sufficient, incl. the background. I made one comment in the attached pdf on literature use in the introduction, where – in chicken – the fermentation of fibre is not the sole function of the caeca (I think fermentation of uric acid is at least also important) and using a citation on a very different species (kiwi) appears inappropriate to me.
The article structure is professional, raw data is provided (albeit in a strange way – split across many different files – there is no reason why the data cannot be provided in a single file in my view, and I would recommend/demand that). This should include both size-specific information per individual (for grit of a certain particle size range) as well as summative information of «all grit».
The manuscript is self-contained with relevant results to hypotheses.

Experimental design

The research is original, and primary. The question is well defined, relevant and meaningful, and the knowledge gap is evident.
The ethical standard is ok, and I guess the investigation methods are ok as well, but the way that the methods are worded precludes a thorough judgement.
In particular, the following method issues are relevant in my view:
The diets need to be described in detail, in terms of composition, and form (pelleted? meal? grain size? etc).
Data that is summative for an individual, such as body weight, gizzard weight, total amount of grit ingested, total amount of grit in the gizzard, total amount of grit in the faeces, should be evaluated by statistical models (either using the raw data, or transformed/ranked data as appropriate) with diet and gizzard muscle state as cofactors, incl. their interaction. This should also be done with relative measures (grit or gizzard weights in % of body mass).
Data on mean grit particle size should be calculated (this is possible, there are methods for this, e.g. in Fritz J, Streich WJ, Schwarm A, Clauss M (2012) Condensing results of wet sieving analyses into a single data: a comparison of methods for particle size description. Journal of Animal Physiology and Animal Nutrition 96:783-797.
ideally, one single value per animal for shape measures would be calculated (using a weighted average approach), and these would be compared between ingested grit, gizzard grit and faecal grit using repeated measures (or, individual as random factor), and diet and gizzard type as covariates.
Some statistical metods and results mentioned in the discussion are never explained in methods or results (see pdf).
The methods of shape need to be explained in detail, ideally with example graphs. Later, in the discussion, «sharpness» is used to describe these measures, but that is not explained in methods. It is also not clear how the measurements in ImageJ were done – how was it achieved that particles did not lie against each other, how were measurements taken (manually, algorithm)?
Sieving was done – and the number of samples for video analysis (n=500 individual stones) was explained for the offered grit. Sieving is not mentioned for gizzard, or faeces, but I assume it was done as well? How many stones did you use for leftover grit/gizzard grit/ faeces grit – 500 per sample as well – or did you measure all particles? How were they arranged for video analysis?
What sieving did you use – wet or dry sieving – what machine was used?
There is a general conceptual issue that influences the methods. Evidently, grit in the gizzard is not only modified in terms of shape, but it is also REDUCED IN SIZE. Hence, even if the ingested grit is never <0.5 mm, the grit in gizzard and especially faeces will definitely contain a lot of material <0.5 mm. How do you deal with this? It seems that this fraction was ignored? That is no problem in terms of methods as long as you discuss it, and the implications. For example, you could calculate the daily ingestion of grit, the daily excretion of grit, and calculate the difference (which, due to your methods, one would have to assume to represent material <0.5 mm). How much would that be? What proportion of the ingested grit would that represent?
I would expect graphical representations of body mass, relative gizzard mass, grit ingestion mass, gizzard grit mass (%BM), mean particle size of ingested/gizzard/faeces grit (e.g. as column charts), i.e. not only charts that represent each individual sieve fraction.
Comparisons should not be made between gizzard/faeces and initial grit, but with ingested grit.
Specific procedures, such as correcting for body mass or using cube root of body mass for relationship with particle size should be stated in the methods.

Validity of the findings

The results are given without selective reporting in my view. Underlying data have been provided. In conclusions, particle size reduction in the gizzard is not mentioned. Most other conclusions are sound, but some are illogical (see pdf).

Additional comments

This is a fascinating study!

---

## Round 0.2 · Major Revisions

Dear authors,

Our reviewers have seen your ms and found several items that need a thorough revision. Revise your ms carefully as the reviewers will see your revision again.

Kind regards
Michael Wink
AE

·

Basic reporting

Thank you for your revision, there definitely has been an improvement in this manuscript, however, I still think it is not ready for publication yet.

Please excuse the delay, but it is holiday and field season right now and due to time constraints I was unable to look at the manuscript with the scrutiny I would have preferred. Especially, I had not the time to check on the statistics. Also, I did not find the figure captions.

I am not a native English speaker, but for me there are still many language issues which need fixing, for example in line 26: How can the ingested grit be sharper (do you mean more angular?) than the offered grit. I thought the ingested grit is part of the offered grit? Do you mean that the proportion of the grit ingested compared to all grit particles that were offered had on average a more angular outline? Also, I would rather limit the use of the term “sharp” here since it is commonly used in connection to sharp edges on tools. Why not use the term “angular”? If you do not wish to use the established geological terminology for roundness, please explain why and at least compare your findings with the six standard categories (Very angular; Angular; Sub-angular; Sub-rounded; Rounded; Well-rounded) at least once.

123: what are silicastic stones? do you mean siliciclastic? if so, do you mean that the stones were composed of siliciclastic sediments (i.e. quartz sandstone) or were they rather composed as one solid vein quartz clast? After looking at fig. S2, I assume that latter, but please be precise as this strongly affects the outcome of your experiments. Also, some clasts on S2 look like chert to me, there might even be a difference in the abrasion rate between vein quartz and chert, at least the development of polish on the surface of chert gastroliths indicate this.

210: I do not understand. 3g were ingested on average, 1g was found in the gizzard, and 2.52g were excreted? where does that additional mass of 0.52g of grit come from? also, with “excreted grit on the last day” you mean the total excreted grit, right?
or do you mean that the combined excreted grit of all chicks on only the last day had this mass? if so, why do you give that amount here when discussing average amounts? and why did you only check the excreted grit mass on the last day?

272 Why did you not check the feces of at least a few chicks during the complete experiment?

294 ability? I am not convinced, they probably just had no need to retain the grit with this diet

301 I am not sure this is entirely true. From hundreds of sets of bird gastroliths I have seen, the fast majority has rather rounded gastroliths. I think it was a rather artificial setting to allow them access to so many angular clasts. This is not what birds usually find in their natural habitats and also not the standard in bird gastroliths. I assume that rounded gastroliths also allow effective trituration of foodstuffs, not by cutting, but by squashing and grinding. Which shape is indeed more effective still needs to be demonstrated.

308-310 Well this is definitely caused by your artificially angular shape of grit particles. Virtually no depositional environment in natural habitats provides birds with such angular stones. I am sure if an ostrich would get access to similarly very angular stones as gastroliths, you would also see an identical very fast change in grit shape. For how fast even the surfaces of quartz gastroliths are altered, see https://www.researchgate.net/publication/307758683_A_simulated_bird_gastric_mill_and_its_implications_for_fossil_gastrolith_authenticity

310-312 I highly doubt that, see above.

319-323 I disagree, as stated above are such angular grit shapes not natural. Also, less muscular gizzards are virtually non-existant in free-ranging herbivorous birds including chicken. For that you would have needed another experimental setting, providing the chicks also access to “normal” rounded grit. All you can say is that in addition to the valid conclusions of Ginofriddo & Best gastroliths shape is also dependent on diet and the strength of gizzard muscles.
Also, if you would have continued your experiment for more than a week (i.e. several months) you would surely have seen that the roundness of the gastroliths would have increased. Not all gastroliths are excreted or replaced within such a short time span.

329 digesting tuff?

Figure S1 – Please crop the images and add a scale bar.
Figure S2 -Please bring all images to the same scale.

Experimental design

see above

Validity of the findings

see above

·

Basic reporting

In my view, this is generally fine. I made some changes in the attached mmanuscript to more accurately reflect the results in my view.
I also make suggestions, both in the word file and in the pdf file, for improvements to the figures. I would include Fig. S2 in the main paper, because it is a very cool and impressive figure - but irrespective of whether it is in supplement or in main document, the scale of all sub-figures needs to be the same, otherwise its effect is lost (right now, it looks as if uningested grit was larger than offered grit, due to the difference in scale).
Supplementary files all need legends in the documents themselves (e.g., all excel tables lack a legend in the excel, and they need units, explanation of abbreviations etc.).

Experimental design

This is fine. There is one thing that is a pity - when the authors counted the 500 pieces of grit from offered or leftover grit, they could have weighed these as well so that they could have made an estimate of the total number of pieces in the offered and uningested grit, which would have allowed them to calculate mean sizes as they did for gizzard and excreted grit. In case this data is available, that should be added and done. But if it is not available, this is not a problem. I added a statement on this in the methods.

Validity of the findings

no comment

Additional comments

please see both the annotated word file (for text and comments on supplements) and the annotated pdf file (for comments on figures and tables of the main text).
This was in my view a very good revision and it is a very nice paper. Thank you for heeding the previous comments so well. In my view, further review is not necessary.
sincerely m clauss

ps: I just noted that I cannot attach the annotated word file, so I will send it via email to the corresponding author and the editor.

---

## Round 0.3 · accepted · Accept

Thank you for the thorough revision. Thus the paper can now be accepted.

Michael Wink
Academic Editor

·

Basic reporting

Thank you for implementing the changes.
I make some additional language and technical remarks in the attached pdf, which should be made at the proofing stage in my view. No further review needed.

Experimental design

fine

Validity of the findings

very good

Additional comments

Thank you for the fine work. m clauss